# A Review on Extracts, Chemical Composition and Product Development of Walnut *Diaphragma Juglandis* Fructus

**DOI:** 10.3390/foods12183379

**Published:** 2023-09-08

**Authors:** Yuanrong Zhan, Mengge Ma, Zhou Chen, Aijin Ma, Siting Li, Junxia Xia, Yingmin Jia

**Affiliations:** 1Beijing Advanced Innovation Center for Food Nutrition and Human Health, School of Food and Health, Beijing Technology and Business University, Beijing 100048, China; 2130022114@st.btbu.edu.cn (Y.Z.); 2250021019@st.btbu.edu.cn (M.M.); zhouch2017@btbu.edu.cn (Z.C.); maaj@btbu.edu.cn (A.M.); lisiting@btbu.edu.cn (S.L.); 2Hebei Yangyuan ZhiHui Beverage Co., Ltd., Hengshui 053000, China; yangyuanshengjibu@hbyangyuan.com; 3Institution of Chinese Walnut Industry, Hengshui 053000, China; 4Hebei Key Laboratory of Walnut Nutritional Function and Processing Technology, Hengshui 053000, China

**Keywords:** walnut, *Diaphragma Juglandis* Fructus, extracts, chemical composition, product development

## Abstract

Walnuts are one of the world’s most important nut species and are popular for their high nutritional value, but the processing of walnuts produces numerous by-products. Among them, *Diaphragma Juglandis* Fructus has attracted the attention of researchers due to its complex chemical composition and diverse bioactivities. However, comprehensive reviews of extract activity and mechanistic studies, chemical composition functionality, and product types are scarce. Therefore, the aim of this review is to analyze the extracts, chemical composition, and product development of *Diaphragma Juglandis* Fructus. Conclusions: For extracts, the biological activities of aqueous and ethanol extracts have been studied more extensively than those of methanol extracts, but almost all of the studies have been based on crude extracts, with fewer explorations of their mechanisms. For chemical composition, the bioactivities of polyphenols and polysaccharides were more intensively studied, while other chemical constituents were at the stage of content determination. For product development, walnuts are mainly used in food and medicine, but the product range is limited. In the future, research on the bioactivity and related mechanisms of *Diaphragma Juglandis* Fructus can be further expanded to improve its value as a potential natural plant resource applied in multiple industries.

## 1. Introduction

The walnut, which belongs to the genus *Juglans*, has been cultivated in more than 50 countries and territories on six continents. Total world walnut production in 2019 reached 4.49 million metric tons. The top five producers are China, the United States, Iran, Turkey and Mexico [1]. China is the world’s largest supplier of walnuts, accounting for more than 40 percent of total global supply, followed by the United States with about 30 percent. China is also the largest consumer of walnuts, with about 81 million tons of shelled walnuts consumed in 2019, twice as much as that consumed in the EU [2]. China’s exports of walnuts (12.9 million tons) were more than 30 times higher than imports (0.4 million tons) in 2022, and total walnut exports (CNY 2.6 billion) accounted for 0.01% of China’s total imports and exports of goods (CNY 23.97 trillion), according to China’s customs data. These figures show that the production and consumption of walnuts in China are extensive and that the development of a healthy walnut industry plays an essential role in the economies of some Chinese provinces.

Walnuts have high economic value and are rich in protein, minerals, fatty acids, phenolic compounds, phytosterols, and dietary fiber. Walnuts consist of walnut green husk, walnut shell, *Diaphragma Juglandis* Fructus (DJF), and walnut kernel (Figure 1), exhibiting health-protective, antibacterial, anti-inflammatory, anti-fungal, and free radical scavenging activities [3]. The walnut kernel is the main part of the walnut consumed due to its high nutritional value, including rich protein and essential unsaturated fatty acids. It can be processed and made into candies, drinks, ice cream, cakes, etc., and is one of the world’s four major nuts [4]. The remaining parts become by-products of walnut kernel processing. Walnut by-products contain a large number of polyphenols, polysaccharides, and other biologically active substances, which have strong development potential for their added value.

DJF, also known as a walnut coat, walnut clip, and walnut septum, is one of the by-products. It is a woody septum within the walnut kernel and accounts for about 4% to 5% of the total mass of walnuts. DJF is brown or light brown, flaky, curved, light, and easily broken [5]. In traditional Chinese medicine, DJF is used to treat insomnia and to nourish the kidneys [6]. In traditional Iranian medicine, it is usually used to treat diabetes [7]. It can also be used to dye hair and fabric in a certain proportion with the outer skin of the pomegranate [8]. These indicate that DJF has promising biological activity. DJF also shows the highest content and species of phenolic compounds among the walnut by-products. For example, Xu [9] compared the polyphenol contents of walnut capsule coat, DJF, walnut green husk, and walnut hull using the Folin–Ciocalteu colorimetric method and found that DJF showed the highest polyphenol content, of 53.89 mg/g. It was also found that the total flavonoid content was highest in DJF (1.00 mg/g), followed in order by walnut capsule rind (0.90 mg/g), walnut green hull (0.45 mg/g), and walnut hull (0.07 mg/g). Zhang et al. [10] compared the phenolic compound species of walnuts, confirming that DJF contained the maximum number of phenolic compound species (41), followed by walnut green husk (32) and walnut flower (29). Additionally, DJF also contains other bioactive components (i.e., saponins, flavonoids, quinones, phenolic acids, esters, diarylheptanes, and polysaccharides) that exhibit favorable biological activities such as anti-tumor, blood glucose regulating, immunomodulating, and sedative and hypnotic effects [11]. Therefore, DJF has the most potential for development. However, there is still room for further research on DJF. With increased consumption of walnuts come increased by-products; the integrated utilization of DJF can provide a more favorable environment for the survival of the walnut industry. This article summarizes the research progress on extracts, chemical composition, and product development of DJF to provide a reference for improving the added value of DJF.

## 2. Extracts

Different extracts have different research focuses. Aqueous extracts are mainly focused on mixtures and their biological activities, ethanol extracts are studied for their biological activities and monomer substances, and methanol extracts are studied for their isolated monomer substances. The biological activity of aqueous and ethanol extracts has been studied more (Table 1). Both have kidney protection and anti-tumor activities, differing in that aqueous extract has anti-fatigue and memory improvement activities and ethanol extract has antioxidant, antimicrobial, hypoglycemic, and sleep improvement activities. Methanol extract has only been found to have anti-tumor and antibacterial activities.

### 2.1. Aqueous Extracts of Diaphragma Juglandis Fructus

The aqueous extract of *Diaphragma Juglandis* Fructus (AeDJF) is mostly extracted by hot infusion, mostly at 85 °C and above, with reflux extraction 2–3 times. The extracts were obtained after concentration and lyophilization. AeDJF has various bioactivities, such as kidney protection, anti-fatigue, memory-improving, and anti-tumor effects (Table 2). These activity studies were based mostly on mixtures rather than monomer substances. Additionally, it was found that the volatile components of AeDJF (i.e., 2,6-dibutyl-4-methylphenol, 3-butyl-4-hydroxyanisole, and 2,3-butanediol) could be used in the food field as antioxidants and food flavoring agents.

AeDJF has significant kidney protection effects. Zhang et al. [12] found that AeDJF had therapeutic effects in hyperuricemic (HUA) mice. Compared with the model group (MG), it could reduce serum uric acid (UA), creatinine (Cr), urea nitrogen (BUN), and liver tissue xanthine oxidase (XOD) activity. It was also able to down-regulate tumor necrosis factor α (TNF-α) and interleukin 1β (IL-1β) levels. Importantly, it could attenuate pathologic structural changes. Zhang et al. [13] also found that in the high-dose group (HDG), AeDJF significantly suppressed XOD activity, regulated oxidative stress, and improved the renal structure in rats with unilateral renal ischemia-reperfusion injury (RIRI). These findings suggest that AeDJF warrants further investigation for therapeutic protection of renal function.

Moreover, AeDJF has anti-fatigue and memory-improving activities. Hou et al. [14] found that AeDJF prolonged the loaded swimming time of mice in a dose-dependent manner. In addition, some researchers investigated the effects of the walnut kernel, walnut inner seed bark, and DJF on mice’s voluntary locomotor exploration ability and memory. The findings suggest that AeDJF can improve memory and regulate blood lipids in mice. Therefore, AeDJF can be further explored and developed for application in memory-enhancing foods [15].

Furthermore, AeDJF can exert anti-tumor effects. Zhu et al. [16] investigated the anti-cervical cancer activity of AeDJF and found that it induced tumor cell necrosis and inhibited sarcoma growth in mice (tumor inhibition rate of 48.80%). The combination of AeDJF with the clinical first-line colon cancer chemotherapeutic drug 5-FU had a synergistic effect on cancer cells, with better results than those obtained with AeDJF alone. Further study found that AeDJF induced apoptosis of HCT-116 human rectal colon cancer cells through endoplasmic reticulum stress [28]. 

Additionally, some volatile components have been separated from AeDJF and could be used in the food and cosmetics fields. For example, Li et al. [29] analyzed the volatile components in AeDJF and concluded that acetic acid, ethyl acetate, 2,6-dibutyl-4-methylphenol, menthol, cedrol, 2,3-butanediol, 1-hexanol, phenethylalcohol, 3-butyl-4-hydroxyanisole, hexanal, and azulene were the main volatile components. A search of the literature revealed that 2,6-dibutyl-4-methylphenol is commonly used as an antioxidant in food processing. Menthol is a natural plant aroma component that is widely used as a freshener and cooling agent in daily products, cosmetics, and pharmaceuticals [30]. Cedarol is a sesquiterpene compound that plays an important role in killing human lung cancer cells [31]. 2,3-Butanediol is widely used in the food field and reacts with acetic acid, producing diacetate-2,3-butanediol ester to exude fruit aroma such as that of melon and banana [32]. 2,3-Butanediol is dehydrogenated to produce diacetyl as a high-value food flavoring agent with certain antibacterial effects [33]. Azulene has antibacterial and analgesic effects, and 3-butyl-4-hydroxyanisole can serve as a fat-soluble antioxidant that is especially suitable for fat-rich foods.

### 2.2. Ethanol Extracts of Diaphragma Juglandis Fructus

The ethanol extract of *Diaphragma Juglandis* Fructus (EeDJF) has been studied more extensively than the aqueous and methanol extracts. The studies were mainly carried out with a focus on the different monomer substances obtained by different volume fractions of ethanol and the extractant, and the biological functions of the extracted mixtures. EeDJF was usually firstly extracted by using ethanol at volume fractions of 60%, 70%, 75%, and 95%. Then, the extracts were further extracted with petroleum ether, ethyl acetate and n-butanol after depressurization and concentration. Next, the small molecule components in the ethanol extract were isolated and purified by using silica gel column chromatography, Sephadex LH-20 gel column chromatography, ODS column chromatography, preparative thin layer, recrystallization, and preparative high-performance liquid chromatography (HPLC). Finally, the monomer compounds were structurally characterized on the basis of physicochemical properties combined with MS, NMR, UV, and IR techniques. 

The ethanol volume fraction and the extractant will have an effect on the resulting monomer substance. A total of 143 substances were isolated and identified (Table 3), among which 71 monomeric compounds were obtained with 70% ethanol extraction (Figure 2A), and 99 monomeric compounds were obtained from the ethyl acetate extraction site. Flavonoids and phenolic acids accounted for the highest percentages (Figure 2B), of 27.27% and 25.87%, respectively. According to Figure 2C, 18 monomers were repeatedly detected at different ethanol concentrations, among which gallic acid and protocatechuic acid were isolated at different extraction ethanol concentrations. Therefore, more small molecule components can be obtained by 70% ethanol extraction followed by ethyl acetate extraction.

With the discovery of various types of small molecules in EeDJF, some researchers have investigated the bioactivities of the extracts. Summarizing the literature, it was found that the biological functions of ethanol extracts mainly include kidney protection, antioxidant, anti-tumor, hypoglycemic, antimicrobial, and other biological functions. 

#### 2.2.1. Protection of the Kidney

Problems with the kidneys can have a huge impact on health. Studies have found that EeDJF can improve kidney function. EeDJF has improved renal function, inhibited XOD activity in liver tissue, and exerted anti-inflammatory effects. UA levels in hyperuricemic mice can be reduced to a certain degree [12]. Zhang et al. [17] found that EeDJF (360 mg/kg) had the effect of increasing oxidase activity, decreasing MDA levels, and lowering TNF-α levels, which could improve acute renal impairment caused by rhabdomyolysis and protect the affected renal tissues. Guan et al. [42] extracted and concentrated DJF with 95% ethanol and obtained extracts of different polarities by sequentially extracting with ethyl acetate and n-butanol. Pharmacological experiments proved that the free radical scavenging capacity, testosterone secretion capacity, and renal function of hydrocortisone-induced kidney Yang deficiency in mice were more significantly improved after administration of the n-butanol part extract. Both the aqueous and ethanol extracts of the DJF are protective of the kidneys, but whether there is a difference in effect between the two has not been assessed. However, in the Ben Cao Gang Mu Shi Yi (Qing Dynasty) and in Xinjiang Uygur traditional medicine, it was mentioned that DJF has a protective function of the kidneys [43]. It has been evident since ancient times that DJF has a favorable effect on the kidneys.

There is no clear conclusion as to the pathway through which DJF extracts exert kidney protection. However, judging from the SOD, MDA, and TNF-α factors affected by AeDJF and EeDJF, the mechanism of action may be to inhibit the inflammatory response, as well as to exert free radical scavenging and anti-lipid peroxidation effects. It is speculated that the related signaling pathways may include the NLR inflammatory body signaling pathway [44,45], ERK1/2 signaling pathway [46,47], NF-κB signaling pathway [48], TLR signaling pathway [49,50], MAPK signaling pathway [51,52], PhoA/ROCK [53], and PI3K/Akt signaling pathway [54,55].

#### 2.2.2. Antioxidant Activity

Free radicals are one or more unpaired electrons containing reactive molecules that damage nucleic acids, proteins, carbohydrates, and lipids, leading to a variety of diseases including early aging, cancer, and atherosclerosis. Antioxidants scavenge these free radicals, preventing cellular damage by ultimately reducing oxidative stress, which can have beneficial effects on human health. The walnut extracts exhibited promising antioxidant activity that was positively correlated with the total phenolic content of walnuts [56]. The 60% EeDJF had a higher total flavonoid content of 21.87 ± 0.55 mg RT/g compared to that of *Penthorum chinense* Pursh, *Artemisia annua* L., and *Citrus aurantium* L [57]. In in vitro assays, n-butanol extracts containing high levels of phenolics and flavonoids could hinder the rancidity and oxidation of rapeseed oil [5]. Similarly, AeDJF and EeDJF exhibited DPPH and ABTS scavenging activity, which was dose-dependent and comparable to that of the positive control (vitamin C, Vc) [18]. Both phenolic acids and flavonoids obtained by isolation had different degrees of DPPH radical scavenging activity [34]. Among them, the phenolic acids such as gallic acid, ethyl gallate, and protocatechuic acid were more active, with half maximal inhibitory concentrations (IC50) of 2.10, 3.55, and 4.99 mg/L (5.96 mg/L for Vc), respectively. The activity of quercetin (IC50 = 3.81 mg/L) and dihydroquercetin (IC50 = 4.82 mg/L) was higher among the flavonoids. The activity of flavonoids such as quercetin (IC50 = 7.98 mg/L) and quercetin-3-*O*-(4″-*O*-acetyl)-α-l-rhamnopyranoside (IC50 = 7.03 mg/L) decreased after glycosylation.

In vivo assays, EeDJF significantly increased antioxidant enzyme activities such as SOD and glutathione (GSH) activity [18]. It reduced ROS accumulation and MDA levels, ameliorated oxidative stress, and prolonged lifespan [19]. The ethyl acetate extract alleviated lipid peroxidation of hepatocyte biofilm and reduced the symptoms of hepatic ischemia-reperfusion injury [58].

The antioxidant activity of phenolic monomer compounds isolated from walnut extract was significantly higher than that of extracts [59]. This could be due to the loss of certain antioxidants during extraction or antagonism among the phenolics. Therefore, research on the antioxidant properties of EeDJF should focus on the isolation of phenolic monomers and systematically explore the synergistic or antagonistic effects among different monomers.

#### 2.2.3. Anti-Tumor Activity

Malignant tumors are one of the greatest causes of human mortality and are one of the most pressing medical challenges to be overcome. The extracts of DJF and some of the isolated monomeric compounds are closely related to anti-tumor effects and thus are presumed to have anti-tumor and immune-enhancing effects. Liu et al. [20] found that EeDJF inhibited HCT116 cell activity and promoted apoptosis in a dose-dependent manner, while also showing the ability to promote the expression of apoptotic factor Bax, decrease the expression of anti-apoptotic factor Bcl2, and promote the cleavage of PARP, a key executive protein of apoptosis. Moreover, it inhibited cell migration and the proliferation of 3D tumor cells. Bai et al. [21] found that EeDJF could increase the apoptosis rate of hepatocellular carcinoma cells by decreasing their proliferation, migration and invasion ability. Meanwhile, it could also alter the expression of proteins related to the Wnt/β-catenin signaling pathway. The mechanism of its action might be associated with the Wnt/β-catenin signaling pathway. Ling [60] extracted DJF with 95% ethanol at reflux and concentrated the ethanol extracts with petroleum ether, chloroform, ethyl acetate, and water-saturated n-butanol in turn. By flow cytometry, it was found that the apoptosis rates of hepatocellular carcinoma cell line 7404, human non-small cell lung cancer cell line A549, human colon cancer cell line Hct116, and Caco-2 cancer cells were all greater than 50% when the concentration of the extracts of DJF ethyl acetate and n-butanol extract sites was 400 μg/mL. Chen et al. [61] demonstrated that ethanol extract of green walnut husks inhibited mTOR phosphorylation and activated the Bcl-2/Bax-dependent apoptotic pathway by regulating the NLRC3/PI3K/AKT signaling pathway, which inhibited colon cancer development. Therefore, we speculate that the anti-tumor effect of EeDJF may be related to certain signaling pathways.

A number of compounds in DJF have anti-tumor activity, such as the quinone compound Juglone, which has inhibitory effects on bladder, cervical, and breast cancer. It may play a role through anti-cell proliferation, induction of apoptosis, induction of autophagy, inhibition of angiogenesis, inhibition of tumor cell invasion and metastasis, induction of DNA damage, inhibition of tumor stem cells, and enhancement of immunity [62]. Ellagic acid, a phenolic compound, can inhibit the proliferation of tumor cells through the VEGFR-2, Notch, PKC, and COX-2 signaling pathways. It also induces apoptosis and blocks epithelial–mesenchymal transition in tumor cells by the PI3K/Akt signaling pathway, JNK (cJun) signaling pathway, mitochondrial pathway, Bcl-2/Bax signaling pathway, and TGF-β/Smad3 signaling pathway. Finally, the MMP SDF1α/CXCR4 signaling pathway inhibits tumor cell metastasis and invasion, induces autophagy, and affects tumor metabolic reprogramming, resulting in anti-tumor effects [63]. β-sitosterol, a phytosterol compound, slowed the proliferation of MCF-7 breast cancer cells and significantly increased the activation of the farnesoid X receptor [64]. As shown in Table 3, these monomers were isolated from EeDJF. The anti-tumor effect of walnuts was found to be an additive or synergistic outcome of several components, including phenolics, phytosterols, tocopherols, and fatty acids, rather than specific components [2]. Therefore, it provides an idea to study the anti-tumor effect exerted by EeDJF. A systematic and efficient analysis of the relationship between mixtures and monomers, with a full consideration of the signaling pathways involved, is necessary.

#### 2.2.4. Hypoglycemic Activity

With the changes in people’s dietary habits nowadays, the prevalence of diabetes is on a continuous increase, causing complications such as heart disease, stroke, diabetic retinopathy, and renal failure, which pose a threat to human health. In Iran, DJF has been traditionally used to treat diabetic patients [7]. One study found that AeDJF and EeDJF modulated blood glucose levels, inhibited liver damage, and attenuated abnormal blood lipid parameters in streptozotocin-induced diabetic mice [22] and alloxan-induced diabetic rats [7], respectively. These suggest that DJF is a potential hypoglycemic agent and to some extent may prevent cardiovascular complications common in diabetic patients. Tan et al. [23] isolated and identified four new taraxasterane-type triterpenes from the 70% ethanol extract and found that all four compounds exhibited a certain degree of α-glucosidase inhibitory activity. Further molecular docking studies revealed that the binding affinity of compound 4 to α-glucosidase was lower than that of compounds 1–3 and the positive control, and compound 4 had lower α-glucosidase inhibitory activity. The in-depth analysis found that the number of hydroxyl groups may be an important influencing factor for the different inhibitory activities of compound 2 and compounds 1, 3, and 4 against α-glucosidase. Tan et al. [35] isolated and identified 11 megastigmanes, 17 flavonoids, and 9 phenylpropanoids from EeDJF. DJF was found to improve diabetic symptoms through the AKT/FoxO1 signaling pathway [65]. Further investigation of compounds 1 to 37 in terms of α-glucosidase inhibitory activity revealed taxifolin, (+)-catechin, quercetin, and luteolin (IC50 values of 40.39, 54.82, 29.47, and 35.41 μM, respectively) exhibited stronger activity than the positive control acarbose (IC50 value of 60.01 μM). From the structural analysis of the compounds, it was found that the structures of the A, B, and C rings in flavonoids were closely related to the inhibitory activity; hydroxylation at the 3-position of flavonoids could enhance the inhibitory effect, saturation of the 2,3-double bond on the C ring could reduce the inhibitory activity [66], the presence of the glycosyl portion on C-3 could reduce the activity [67], and the gallic alcohol portion could enhance the inhibitory activity of flavonoids [68].

#### 2.2.5. Antibacterial Properties

Due to the increasing severity of bacterial resistance, drug-resistant bacteria have become a major problem threatening human health, and there is an urgent need to find new therapeutic drugs. Numerous studies have shown that compounds extracted from plants, animals, and microorganisms have antibacterial activity and are expected to be novel antibacterial agents. Extensive experiments have demonstrated the antibacterial effects of walnut extract [69,70,71,72,73]. In fact, EeDJF has promising antibacterial activity. Gao et al. [74] found that EeDJF had more significant antibacterial activity than the aqueous extract against *Bacillus subtilis*, *Staphylococcus aureus*, *Escherichia coli*, and *Bacillus aerogenes*. Ethyl acetate [24] and n-butanol [25] extracts exhibited promising bacteriostatic effects. Among the phenolic compounds, phenolic acids showed the strongest antibacterial activity, especially ethyl gallate, followed by flavonoids and naphthol glycosides, and the weakest were dihydrotetrasol glycosides. The antibacterial activity of flavonoids before and after glycosylation was found to differ. The antibacterial spectrum of phenol glycosylation was narrower than that of aldehydes. Additionally, structurally similar compounds did not necessarily have similar antibacterial activity. The same compounds also did not inhibit the growth of all bacteria with the same properties. These findings suggest that the antibacterial activity of compounds is related not only to their structures and bacterial properties but also to additional influencing factors that need to be further investigated in order to accurately grasp the mechanism of their antibacterial activity [25].2.2.6. Other Biological Functions

EeDJF has also been shown to have sleep-improving and anti-inflammatory activities. Zhang et al. [26] found that the combination of EeDJF and pentobarbital significantly shortened sleep latency, prolonged sleep duration, and improved sleep incidence in mice. Wang et al. [41] isolated 14 compounds from the ethyl acetate extraction site after extracting the DJF with 95% ethanol. The compounds such as gallic acid, ethyl gallate, and (+)-dehydrovomitol exhibited inhibitory activity in an in vitro RAW 264.7 macrophage model stimulated by lipopolysaccharide. Zhao et al. [36] found for the first time both α-D and α-L conformations of taxifolin-3-O-arabinofuranosides in walnut. This suggests that the flavonoids present in the DJF are a special form and may have potential biological activity.

### 2.3. Methanol Extracts of Diaphragma Juglandis Fructus

Fewer studies have been carried out on the methanolic extracts of *Diaphragma Juglandis* Fructus (MeDJF). The monomer substances obtained from methanol extraction were mainly ketones and lignans, which were investigated for their anti-tumor and antimicrobial functions. MeDJF showed in vitro inhibitory activity against cervical cancer cell (Hela), human gastric cancer cell (HGC-27), and colon cancer cell (Ht-29) lines. The substances that acted were hexahydro-demethoxycurcumin-A, juglanin B [75], 2,3-dihydroxy-1-(4′-hydroxy-3′-methoxy-phenyl)-propan-1-one, 3′,4′-dimethoxyphenylpropanediol, (2S)-3,3-di-(4-hydroxy-3-methoxyphenyl)-propane-1,2-diol, and (7S,8R)-dihydrodehydrodiconiferyl alcohol [27]. Most of them belong to lignans and ketones. Additionally, the antibacterial activity of methanol extract was stronger than that of aqueous and ethanol extracts [8]. Numerous researchers have found that methanol extracts have better antibacterial activity. For example, the methanol extract of Anamur banana (*Musa Cavendishii*) was the most active in all enzyme inhibition and antimicrobial and antioxidant activity assays, mainly due to its abundant phenolic content [76]. In addition, the methanol extract of *Origanum haussknechtii* showed the highest antimicrobial activity (MIC of 100 µg/mL) against *Escherichia coli* [77]. Rajendrasozhan et al. [78] also found that an 80% methanolic extract of *Rhanterium epapposum* showed potent antifungal activity against all fungal strains tested. Therefore, research on the antibacterial activity of MeDJF can be intensified, and its main antibacterial substances can be analyzed to provide theoretical references in the field of antibacterial agents.

In conclusion, for isolation and purification of monomer substances, ethanol extracts are more studied, followed by methanol extracts, and aqueous extracts are hardly involved. Regarding biological activity, aqueous and ethanol extracts were studied more than methanol extracts (Table 1). Analysis revealed that further studies on the mechanisms of bioactivity of these three extracts are needed. Therefore, it will be worthwhile to continue to explore new biological activities and mechanisms of action in the future.

## 3. Chemical Composition

Modern research findings indicate that DJF contains polysaccharides, flavonoids, phenols, saponins, crude fat, crude protein, amino acids, moisture, total alkaloids, and crude fiber, with contents of 2.48–5.40% [8], 2.54–9.09% [8], 2.39–3.54% [8], 1.86–5.98% [8], 0.01–3.12% [8], 1.09–4.20% [8], 1.00–3.21% [8], 14.75% [6], 3.81% [79], and 30.82% [80], respectively. From the separation and purification of different monomers obtained from ethanol and methanol extracts, different chemical compositions in DJF had been demonstrated, with polyphenols being the dominant substance. There have been more studies on the polyphenol and polysaccharide activities of DJF.

### 3.1. Polyphenols

Polyphenols in walnuts have various physiological functions such as antioxidant, anti-inflammatory, anti-tumor, and cardiovascular-protective activities [81]. DJF is more valuable in terms of polyphenol and flavonoid content among walnut by-products. Liu et al. [81] detected 200 compounds from DJF, with dietary polyphenols such as hydrolyzed tannins, flavonoids, and phenolic acids accounting for the majority. Further studies found that the ellagic acid content (518.38–1733.64 μg/g) in DJF was considerable and comparable to or even higher than that in blackberries, cloudberries, strawberries, raspberries, fruit juices, and walnut kernels. The contents of (+)-catechin (251.69–693.32 μg/g), (-)-epicatechin (8.82–36.44 μg/g) and (-)-epicatechin gallate (22.30–194.79 μg/g) were significantly higher than those in walnut kernels (14.80–82.00, 0.34–1.49, and 2.72–13.22 μg/g, respectively). Hu et al. [6] found that the main phenolic compounds in DJF existed in their free form, with abundant types. A total of 11 free phenolic substances and 10 combined phenolic substances were detected in DJF, indicating that DJF could be used to develop functional foods beneficial to health, such as phenolic antioxidants, regulators of intestinal ecology, weight loss aids, and iron supplements.

Polyphenol research focuses on the extraction, purification, and bioactivity of polyphenols. The extraction of polyphenols from DJF is mostly performed by ultrasound-assisted techniques and enzyme-assisted techniques (Table 4). For extraction yield, the highest polyphenol content of 63.92 mg/g was obtained by the synergistic action of enzymes and ultrasound [82]. Li et al. [83] found that the order of factors influencing the extraction of polyphenols from Yunnan walnut DJF was water bath temperature > material-to-liquid ratio > complex enzyme addition amount. After purification with AB-8 macroporous resin, the polyphenol purity increased from 19.74% to 62.09%. Chen et al. [84] detected 12 polyphenols by the UPLC method, among which rutin had the highest content (1168.78 μg/g DW), followed by isorhamnetin (713.85 μg/g DW), with lower content of syringic acid (124.63 μg/g DW) and p-coumaric acid (124.77 μg/g DW). The antioxidant activity of polyphenols was investigated by the FRAP, DPPH, and ABTS methods and was found to be higher than those of fruits [85,86], wild flowers [87], vegetables [88], and cereals [89].

Flavonoids are a general term for a series of substances, mainly α-phenylbenzopyrones, that have wide applications in disease treatment and food additives. They can not only inhibit the production of various free radicals but also remove excessive free radicals in the body. Thus, they indirectly play a role in delaying aging, preventing cardiovascular and cerebrovascular diseases, and preventing cancer [90]. The ultrasonic-assisted alkaline technique and ultra-high-pressure technique yielded higher total flavonoids from DJF, of 13.95% and 12.31%, respectively (Table 4). Flavonoids have good antioxidant activity, but this can differ depending on the extraction method. For example, Zhao et al. [91] obtained total flavonoids extracted with 55% ethanol showing maximum •OH and O^2−^• scavenging activity of 88.04% and 89.56%, respectively, at a concentration of 1 mg/mL. However, flavonoids obtained by Zhao et al. [92] utilizing the ultrasound-microwave technique had maximum •OH and O^2−^• scavenging activity of only 66.20% and 36.90%, respectively, at a concentration of 2 mg/mL. Some researchers further analyzed the results of antioxidant capacity measurements. They found that the structures with B-4′-OH, B-3′-OH, A-7-OH, B-ring with an adventitious phenolic hydroxyl group, 3′,4′-*o*-dihydroxy, C_3_-OH, C_2_=C_3,_ and C-4 carbonyl groups can provide hydrogen donor and oxygen donor protons to block the chain reaction of free radicals [90,93,94]. This enhances the antioxidant activity of flavonoids.

Flavonoids have hypoglycemic activity. Li et al. [65] screened 10 potentially active ingredients and 15 potential targets by network pharmacology. Flavonoids such as rutin and quercetin in DJF were found to have anti-diabetic effects by increasing AKT protein phosphorylation and decreasing FoxO1 protein expression in HepG2 cells. Additionally, the total flavonoids in DJF significantly, competitively, and reversibly inhibited the enzymatic activities of α-amylase and α-*D*-glucosidase [95] and could be developed as therapeutic agents for diabetes to increase DJF’s value.

Liu et al. [96] extracted and determined phenolic acids from different parts of walnuts and found that the phenolic acid content of DJF was second only to that of walnut seed coat. Meanwhile, the content of gallic acid was the highest in DJF from different origins, followed by caffeic acid and methyl gallate. In vitro and in vivo experiments confirmed the high antioxidant capacity of phenolic acids in DJF. The antioxidant activity of phenolic acids was stronger than that of flavonoids [34].

**Table 4 foods-12-03379-t004:** Summary of extraction of polyphenolic substances from DJF.

Name	Extraction and Purification Conditions	Yield/Extraction Amount	Monomeric Substances	References
Polyphenols	Enzyme-assisted technology: material-to-liquid ratio of 102.00 mL/g, complex enzyme addition (pectinase and cellulase) of 0.90%, temperature of 45.33 °C. Purification: AB-8 macroporous resin, 90% ethanol elution.	22.29 mg GAE/g; the polyphenol purity changed from 19.74% to 62.09%.	N/A	[83]
Polyphenols	Ultrasonic-assisted technology: material-to-liquid ratio of 1:62 (g/mL), volume fraction of ethanol of 50%, ultrasonic time of 50 min, ultrasonic temperature of 71 °C.	6.98%.	N/A	[97]
Polyphenols	Ultrasonic-assisted technique: material-to-liquid ratio of 1:40 (g/mL), ethanol concentration of 30%, ultrasound for 15 min, extraction temperature of 50 °C.	56.46 mg GAE/g.	rutin, isorhamnetin, syringic acid, and p-coumaric acid.	[84]
Polyphenols and Flavonoids	Ultrasonic cellulase simultaneous extraction method: enzyme mass fraction of 0.4%, material-to-liquid ratio of 1:70 (g/mL), 50% ethanol, pH = 4.8, ultrasonic power of 210 W, extraction temperature of 50 °C, extraction time of 40 min, 2 times extraction.	Flavonoids (121.36 mg/g); polyphenols (63.92 mg/g).	N/A	[82]
Flavonoids	Ultrahigh-pressure extraction: ethanol volume fraction of 62%, pressure of 385 MPa, holding time of 9 min, material-to-liquid ratio of 1:25 (g/mL).	12.31%.	N/A	[98]
Flavonoids	Material-to-liquid ratio of 1:13, 59% ethanol, 2 times of reflux extraction, 77 min each time.	64.12 mg/g.	N/A	[95]
Flavonoids	Ultrasonic-assisted technology: 60% ethanol, material-to-liquid ratio of 1:30 (g/mL), ultrasonic power of 200 W, ultrasonic extraction time of 1 h.	7.76%.	naringin, rutin, isoquercetin, hyperoside, dihydroquercetin, catechin, quercitin, gallic acid, quercetin, astragaloside, epicatechin gallate.	[90]
Flavonoids	Ultrasonic-assisted lye technology: lye concentration of 0.4 mol/L, ultrasonic time of 1.5 h, material-to-liquid ratio of 1:37.	13.95%.	N/A	[99]
Flavonoids	Ethanol concentration of 55%, extraction temperature of 80 °C, extraction time of 80 min.	9.88%.	N/A	[91]
Flavonoids	Ultrasonic-microwave synergistic technique: ethanol concentration of 50%, material-liquid ratio of 1:25.6 (g/mL), ultrasonic power of 261.8 W, microwave power of 150 W, extraction temperature of 51.6 °C, extraction time of 8.5 min.	5.17%.	N/A	[92]
Total phenolic acid	Ethanol concentration of 40%, extraction temperature of 80 °C, extraction time of 60 min, material-to-liquid ratio of 1:21 (g/mL).	N/A	gallic acid, caffeic acid, protocatechuic acid, methyl gallate.	[96]

### 3.2. Polysaccharides

Polysaccharides are important chemical constituents in DJF, which contains concentrations of 2.48% to 5.40% [19]. To date, the polysaccharide extraction methods, monosaccharide composition and molar ratio, molecular weight, and biological activity of the polysaccharides in DJF have been studied. Polysaccharides in DJF are mostly extracted by hot water extraction and purified by ion exchange chromatography and gel chromatography (Table 5). Hu et al. [6] analyzed DJF with HPLC and found that it contained nine monosaccharides, such as xylose, trehalose, and mannose (Table 5). The monosaccharide content was 314.16 mg/mL. The highest content of trehalose was 223.76 mg/mL, which was close to the monosaccharide content of *Cordyceps militaris* (L.) (247.1 mg/g) [100]. Other monosaccharides found in the polysaccharides of DJF were fructose, glucose, and fucose (Table 6). It was also found that monosaccharide molar ratios differed when the monosaccharide composition of DJF was consistent. The molecular weight of polysaccharides ranged from 3000 Da to 13,000 Da, probably due to the different origins and species of walnut species.

Polysaccharides were proven to have antioxidant, antibacterial, anti-tumor, immunomodulatory, hypoglycemic, and anti-late glycosylation end-product formation activities. The water-soluble polysaccharides DJP-2 [101] and SJP-2 [102] showed significant antioxidant activity in vitro. The results of pharmacological studies showed that P5a inhibited the formation of advanced glycosylation end-products (AGEs) more than P4a [25]. DJP-2 had inhibitory effects on Amadori products at the early stage of AGE formation, on dicarbonyl compounds at the middle stage, and on fluorescent AGEs at the late stage, which could lead to applications for DJP-2 in the treatment of various diseases related to oxidative stress and AGEs [103]. A continued study of DJP-2 revealed that it could significantly inhibit the growth of Gram-negative and Gram-positive bacteria in a dose-dependent manner [101]. It was shown that DJP-2 could effectively inhibit the proliferation of HepG2 and BGC-82 cell lines by enhancing phagocytosis, stimulating the production of NO, tumor necrosis factor-α (TNF-α), and interleukins (IL-6 and IL-1β), and promoting the expression of their corresponding mRNAs in a dose-dependent manner. Meanwhile, DJP-2 could be a novel anti-tumor and immunomodulatory agent, as CR3, MR, and TLR2 were shown to be the major membrane receptors for DJP-2 on RAW 264.7 [104]. Further, DJP-2 had significant hemolysis inhibitory activity and effectively attenuated oxidative damage to hepatic L02 cells by H_2_O_2_ by enhancing cell viability. In vitro, DJP-2 dose-dependently inhibited the enzymatic activity of α-amylase and α-*D*-glucosidase and reduced blood glucose in streptozotocin-induced diabetic mice. Both in vitro and in vivo experiments have shown that DJP-2 has a good regulatory effect on postprandial hyperglycemia [103]. DJP-2 shows that the same DJF polysaccharide has various biological functions and can be applied in different fields.

**Table 5 foods-12-03379-t005:** Types and contents of monosaccharides in DJF.

Number	Identity	Amount/(mg/g)	Molecular Weight	Formula	Structural Formula	References
1	Mannose	11.45 ± 0.52	182.17	C_6_H_14_O_6_	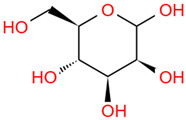	[6]
2	Rhamnose	8.99 ± 0.41	164.16	C_6_H_12_O_5_	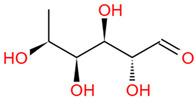	
3	Ribose	2.99 ± 0.27	150.13	C_5_H_10_O_5_	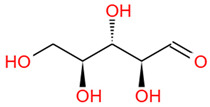	
4	Glucuronic acid	3.09 ± 0.09	194.14	C_6_H_10_O_7_	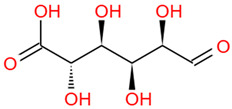	
5	Trehalose	223.76 ± 10.01	342.30	C_12_H_22_O_11_	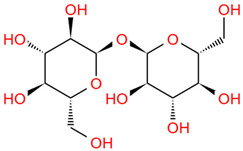	
6	Galacturonic acid	8.18 ± 0.05	194.14	C_6_H_10_O_7_	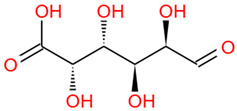	
7	Xylose	44.79 ± 2.08	150.13	C_5_H_10_O_5_	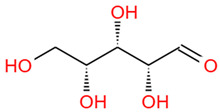	
8	Galactose	2.77 ± 0.02	180.16	C_6_H_12_O_6_	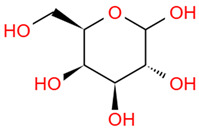	
9	Arabinose	8.11 ± 1.57	150.13	C_5_H_10_O_5_	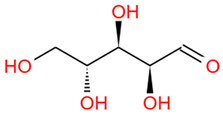	

**Table 6 foods-12-03379-t006:** Polysaccharides of DJF.

Name of Polysaccharide	Extraction and Purification Methods	Monosaccharide Composition and Molar Ratio	Molecular Weight	Biological Activity	References
Acid polysaccharide SJP-2	Crushed through 20 mesh sieve, material-to-liquid ratio of 1:20 (*m*/*v*), extraction at 90 °C for 2 h, concentrated and lyophilized, depigmented with hydrogen peroxide, concentrated, and deproteinated by Sevage method. Purification: DEAE-Sepharose Fast Flow ion exchange column (2.0 cm × 35 cm) and Sephadex-G50 gel column chromatography (1.0 × 64 cm).	arabinose:galactose:glucose:galacturonic acid = 1:2.62:4.45:18.53.	3239 Da	Antioxidant. The IC50 values for scavenging DPPH radicals, hydroxyl radicals, and ABTS radicals were 0.094, 0.945, and 0.591 mg/mL, respectively, and the reducing capacity reached 1.158 at 1.0 mg/mL. The ORAC value was 1217.99 ± 31.22 μmol TE/g.	[102]
P4a and P5a	Crushed and sieved through 60 mesh sieve at a ratio of 1:40 (*m*/*v*), extracted 3 times at 85 °C for 3 h. Concentrated by filtration, alcoholic sedimentation, centrifuged, washed with anhydrous ethanol, sediment spread out, and evaporated in a ventilated area until no alcoholic odor was present. Purification: DEAE-Sephadex A-25 anion-exchange ion column chromatography (2.6 × 46 cm) and Sephadex G-75 gel column chromatography (2.6 × 96 cm).	P4a: mannose:rhamnose:glucose:galactose:xylose:arabinose:fucose = 1:23.1:11.4:72.2:19.9:25.7:2.6. P5a: mannose:rhamnose:glucose:galactose:xylose:arabinose:fucose = 1:36.2:26.7:82.7:12.3:21.9:2.5.	P4a:7015 Da. P5a:13057 Da.	Inhibition of α-glucosidase and AGE formation.	[25]
Water-soluble polysaccharide DJP-2	Raw material ratio of 20 mL/g, extraction time of 40 min, and microwave extraction power of 400 W. Purification: DEAE-Cellulose 52 ion exchange column (2.6 × 60 cm) and Sephadex G-100 gel column (2.6 × 60 cm).	Arabinose:galactose:glucose:xylose:mannose = 0.27:0.55:1.00:0.14:0.08.	4950 Da.	Antioxidant and antibacterial. Antitumor and immunomodulation. Mitigation of oxidative damage to hepatic L02 cells by H_2_O_2_. Blood glucose regulation: inhibits α-amylase and α-*D*-glucosidase activity, lowers blood glucose, inhibits AGEs formation.	[101,103,104]

### 3.3. Saponins

Studies have demonstrated that saponins have biological effects and medicinal properties, such as anti-inflammatory, antibacterial, antiviral, insecticidal, anticancer, and cholesterol-lowering activity [105], with favorable application prospects. Studies on DJF saponins are still in the extraction and purification stage. The total saponins yield ranges from 1.86% to 4.19% (Table 7). The extraction of saponins from DJF was mostly performed with different volume fractions of ethanol, with a material-to-liquid ratio of 1:25, reflux extraction at 80 °C, and also with the help of ultrasonic-assisted extraction. The factors affecting the yield of total saponins were in the order of ethanol concentration > material-to-liquid ratio > extraction time [106]. Purification of crude saponins could be achieved with D101 macroporous adsorbent resin, and the purity of total saponins increased by 23.40%. It was found that total saponins of DJF had better scavenging ability against DPPH radicals, hydroxyl radicals, and superoxide anion radicals in a dose-dependent manner, but were basically weaker than Vc [60].

### 3.4. Fatty Acids

The content and variety of unsaturated fatty acids and saturated fatty acids are rich in DJF. Palmitic acid (700.55 ± 8.80 mg/kg) was the highest among saturated fatty acids and is a commonly used natural emulsifier in food and cosmetics. Oleic acid (549.92 ± 18.98 mg/kg) and linoleic acid (1314.06 ± 10.71 mg/kg) were the dominant types of unsaturated fatty acids. DJF also contained minor amounts of polyunsaturated fatty acids such as eicosapentaenoic acid, Arachidonic acid, and neuronic acid, which have been used as health food ingredients in fortified foods [6].

### 3.5. Amino Acids

Essential and non-essential amino acids in DJF are more abundant in both species and content. Hu et al. [6] detected six essential amino acids (EAAs) and 11 non-essential amino acids (NEAAs) in DJF and walnut shells. They found that the contents of EAAs and NEAAs decreased sequentially in walnut kernels, DJF and walnut shells. The EAAs with the highest contents in DJF were lysine (2.41 mg/g dw) and leucine (1.53 mg/g dw).

### 3.6. Metal Elements

DJF might be a potential source of iron for the human body because its iron content is higher than that of Turkish walnut [109] and other nuts reported in the literature [97]. Hu et al. [6] detected the contents of K, Na, Ca, Mg, Fe, Cu, Zn, Mn, and Se in DJF as 8362.87 mg/kg, 479.17 mg/kg, 3526.37 mg/kg, 636.80 mg/kg, 36.93 mg/kg, 3.78 mg/kg, 4.93 mg/kg, 14.70 mg/kg, and 0.05 mg/kg, respectively.

In summary, among the chemical constituents of DJF, polysaccharides and polyphenols have been more extensively studied, while all other constituents have only been determined in terms of their contents. Further analysis revealed that the polysaccharide structure studies were mainly aimed at determining the molecular weight, monosaccharide composition, and microstructure, with less research on glycosidic bond types and other higher structures. The types and contents of DJF polyphenols were absolutely dominant in walnut by-products, so they were studied in detail. It was found that the content and structure of polyphenol monomer substances were related to antioxidant capacity, as the number and position of hydroxyl groups, the carbonyl structure, and the position of glycosidic bonds would all affect the antioxidant capacity. The presence of these components indicates that DJF has high added value as a walnut by-product and is likely to be applied in the food field and other industries.

## 4. Product Development of Diaphragma Juglandis Fructus

DJF has biological functions such as kidney protection, memory improvement, and anti-fatigue effects. But, there are relatively few products made from it as a raw material. In order to improve the utilization of DJF, many researchers have developed different products (Table 8), including food products such as tea beverages, bagged teas, coloring agents, herbal formulations in medicine, and others.

In the food field, DJF has been developed for tea beverages, bagged teas, effervescent tablets, fruit wines, and food additives. Using DJF, licorice, and honeysuckle as raw materials, a compound herbal tea beverage with golden color, stable quality, and unique flavor was obtained by double enzymatic digestion [110]. Bagged tea is popular among many people due to its convenience and health effects. Liu [111] analyzed a brewing solution of DJF bagged tea. It had aqueous extract content of 68.60%, total flavonoid content of 5.62%, and polysaccharide content of 4.74%, and 27 flavor substances were also detected. Effervescent tablets are gradually gaining popularity because of their small size, light weight, easy portability, nutritional richness, and easy, effective absorption by the body. Some researchers used processed jujube pomace as a raw material and compounded it with DJF powder to make functional compound effervescent tablets [112]. Fruit wines, coloring agents, and preservatives have been developed based on the nutritional composition of DJF. Chen et al. [113] used walnut flowers, walnut leaves, walnut shells, and DJF as raw materials to make fruit wine with a unique flavor and excellent taste. Luo [114] obtained a colorant with good colorant effect, stable performance, and no harmful substances by using ultrasonic extraction and freeze spray drying technology with DJF as raw material. A sheep milk and lamb preservative with favorable edible taste and antioxidant and antimicrobial properties was developed based on the characteristics of DJF, buckwheat bran, and raisins for dual use in medicine and food [115].

In the medicine field, DJF has been used to research treatments for several diseases. Zhao et al. [116] obtained a phenolic extract of DJF with a flavonoid content of 64.50 to 66.10%. They found that it significantly inhibited various assays in animals with abnormal glucolipid metabolism at both low and high doses. Thus, it could be used as a food or feed additive and raw material for human or veterinary drugs. Jiang et al. [117] studied an AeDJF that can be used for the treatment of ulcerative colitis. It can effectively treat and alleviate the symptoms of ulcerative colitis, such as weight loss, diarrhea, and blood in stool. Moreover, it can regulate the composition of intestinal flora and promote flora recovery. Cong [118] invented a combination Chinese herbal formula based on *Alpiniae Oxyphyllae* Fructus, *Semen Platycladi*, and DJF with promising efficacy for the treatment of Parkinson’s disease of the type with kidney Yang deficiency, liver heat, blood, and wind. Additionally, it can be used to tonify liver blood and remove liver wind without side effects. Shen et al. [119] observed the effect of DJF combined with prostatic artery embolization in the treatment of benign prostatic hyperplasia. The patient showed significant improvement in prostatic symptoms without serious complications after 7 days of continuous treatment with DJF.

In other fields, nutrients for crops, activated charcoal, biomass charcoal, and cosmetics have been developed on the basis of the physical properties of DJF. Wang et al. [120] used DJF as a nutrient component for cultivating white mushrooms and obtained white mushrooms with high yield, large bodies, rich flesh, uniform thickness, high-temperature resistance, and high nutritional value. Yan [121] used DJF as a biomass carbon doped into a precursor for the preparation of composite photocatalysts, thus increasing the degradation efficiency of the composites, while the catalytic performance was basically unchanged after five cycles of recycling. Qian [122] invented an all-natural skin care combination with melanin reduction and whitening effect, in which an extract of DJF could accelerate the production and excretion of melanin from the skin at the cellular level to solve the whitening problem at the root.

Although DJF product development has been pursued in food, medicine, and other areas, the range of products remains small, and no product with distinctive advantages has yet been developed. In the food field, as a raw material with rich biological activity, DJF still needs to be further developed into a listed product with unique advantages. In the medicine field, due to its long history of medicinal use, it needs to be developed into a targeted traditional Chinese medicine formula. In other areas, the development of biomass charcoal, crop nutrients, and cosmetics has shown that DJF can be applied in various industries such as new materials, agriculture, and cosmetology. Further research is warranted for the development of different application areas.

**Table 8 foods-12-03379-t008:** Product development of DJF.

Category	Product	Raw Materials	Features	References
Food	A compound herbal tea beverage	DJF, licorice, and honeysuckle.	Golden color, stable quality and unique flavor were obtained by double enzymatic digestion.	[110]
	bagged tea	DJF.	Its aqueous extract content was 68.60%, total flavonoid content was 5.62%, polysaccharide content was 4.74%, and 27 flavor substances were detected. The flavor substances with higher content were linalool (4.04%), limonene (3.41%), γ-terpene (1.54%), lauricene (1.82%), and furfural (1.05%).	[111]
	Effervescent tablets	An 8:2 ratio of jujube pomace to DJF, 0.75% meringue addition, 0.8% anhydrous citric acid addition and 0.25% silicon dioxide addition.	The theoretical shelf-life was 68 days at 25 °C and 60% humidity.	[112]
	Fruit wine	Walnut flowers, walnut leaves, walnut shells, and DJF.	A unique flavor and excellent taste.	[113]
	Colorant	DJF.	Good colorant effect, stable performance, and no harmful substances.	[114]
	A sheep milk and lamb preservative	DJF, buckwheat bran, and raisins.	Favorable taste, antioxidant, and antimicrobial properties.	[115]
Medicine	The treatment of ulcerative colitis	AeDJF.	It can effectively treat and alleviate the symptoms of ulcerative colitis, such as weight loss, diarrhea, and blood in stool. Additionally, it can regulate the composition of intestinal flora and promote flora recovery.	[117]
	A combination Chinese herbal formula for the treatment of Parkinson’s disease	Alpiniae Oxyphyllae Fructus, Semen Platycladi, and DJF.	It has promising efficacy in Parkinson’s disease of the type with kidney Yang deficiency, liver heat, blood, and wind. Also, it can be used to tonify liver blood and remove liver wind without side effects.	[118]
	The treatment of benign prostatic hyperplasia	DJF combined with prostatic artery embolization.	The patient showed significant improvement in prostatic symptoms without serious complications after 7 days of continuous use with DJF.	[119]
Others	A nutrient component for cultivating white mushrooms	DJF.	Obtained white mushrooms with high yield, large bodies, rich flesh, uniform thickness, high-temperature resistance, and high nutritional value.	[120]
	Composite photocatalysts	DJF as a biomass carbon doped into the precursor.	Increased the degradation efficiency of the composites, and the catalytic performance was basically unchanged after five cycles of recycling.	[121]
	An all-natural skin care combination	Mung bean germ extract, Bletilla striata extract, mulberry leaf extract, DJF extract, etc.	Melanin reduction and whitening effect. The extract of DJF could accelerate the production and excretion of melanin from the skin at the cellular level to solve the whitening problem at the root.	[122]

## 5. Conclusions and Future Perspectives

Nowadays, more and more people are pursuing the original concept of healthy living, so the development of bioactive ingredients in natural products is becoming more and more urgent. The summary of the extracts and chemical composition of DJF shows that DJF has multiple active ingredients, such as polysaccharides and polyphenols. These substances have antioxidant, antibacterial, hypoglycemic, kidney protection, anti-tumor and sleep-enhancing effects. DJF is very suitable for people’s current needs and has a wide development prospects. Therefore, it can be concluded that the comprehensive utilization of DJF will bring new opportunities to the walnut industry. It will also provide new products in functional food, medicine, and other fields.

Although bioactivity studies have been conducted on the extracts and chemical composition of DJF, they are still at a preliminary stage. The following points may be future directions for research and development:Isolation and structural identification of the monomeric substances in the extracts, especially for the structural characterization of homogeneous polysaccharides.Systematic study of synergistic or antagonistic effects among monomers. Comprehensive analysis of the differences in the effects of monomers and mixtures on specific biological activities.Flavonoids and other substances have strong antioxidant activity. The study of the relationship between the structural characteristics of the monomers and their biological activities can be effective in obtaining components with better effects.Based on the biological activity of DJF, continue exploring and developing products that are popular among and beneficial to consumers.

## Figures and Tables

**Figure 1 foods-12-03379-f001:**
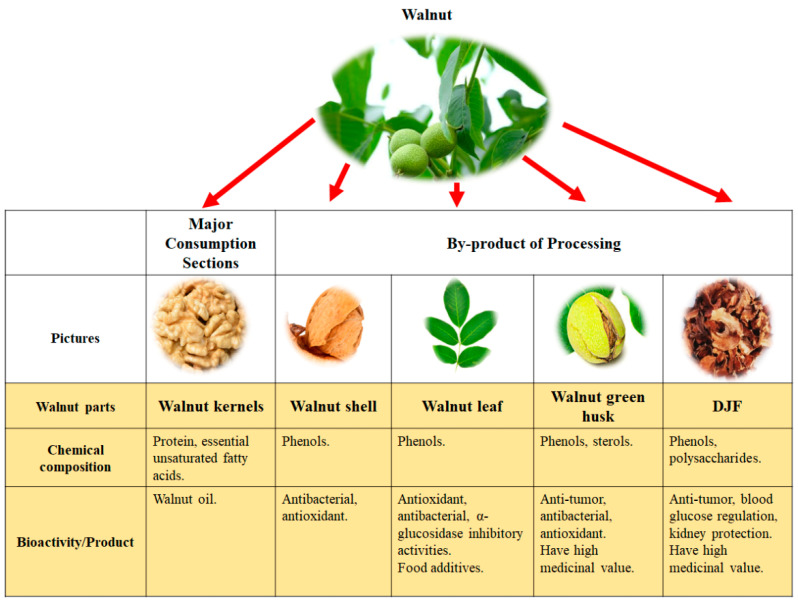
Bioactive components and functions of each part of walnut.

**Figure 2 foods-12-03379-f002:**
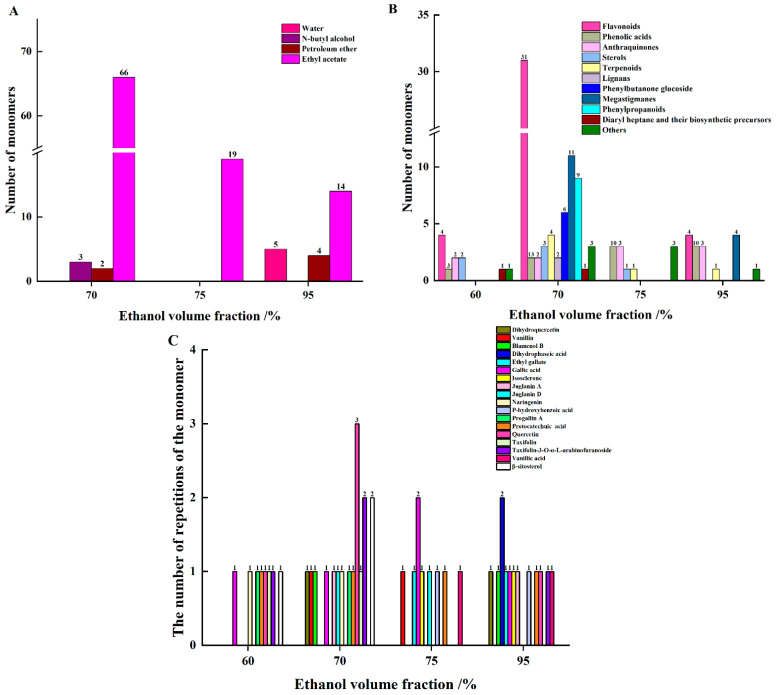
Number of monomers (**A**), monomer classification (**B**), and monomer duplication (**C**) obtained from different extraction sites at different ethanol volume fractions.

**Table 1 foods-12-03379-t001:** Summary of biofunctions of different solvent extracts of *Diaphragma Juglandis* Fructus (DJF).

Extracts	Biological Activities	Effects	References
Aqueous extracts	Kidney protection	Reduced serum uric acid, creatinine, and urea nitrogen. Reduced activity of xanthine oxidase in liver tissue. Reduced IL-1β and TNF-α. Regulated oxidative stress.	[12,13]
	Anti-fatigue	Prolonged the time to exhaustion.	[14]
	Memory improvement	Improvement in spontaneous exploratory ability and memory.	[15]
	Anti-tumor	Caused tumor cell necrosis. Inhibited sarcoma growth.	[16]
Ethanol extracts	Kidney protection	Increased oxidase activity. Improved free radical scavenging capacity and testosterone secretion capacity. Reduced malondialdehyde. Reduced TNF-α at the 48th hour.	[12,17]
	Antioxidation	Scavenged DPPH and ABTS. Improved antioxidant enzyme activities.	[18,19]
	Anti-tumor	Enhanced the apoptosis rate. Reduced proliferation, migration, and invasion ability. Wnt/β-catenin signaling pathway.	[20,21]
	Hypoglycemia	Inhibited liver damage. Reduced abnormal blood lipid parameters and α-glucosidase activity.	[7,22,23]
	Antibacterial	Ethyl acetate and n-butanol site extract showed favorable inhibitory effects.	[24,25]
	Sleep improvement	Shortened sleep latency. Prolonged sleep duration. Improved sleep incidence.	[26]
Methanol extracts	Anti-tumor	Inhibited Hela, HGC-27, and Ht-29.	[27]
	Antibacterial	The antibacterial activity of methanol extract was stronger than that of aqueous or ethanol extract.	[8]

**Table 2 foods-12-03379-t002:** Biological activities of aqueous extracts of DJF.

Biological Activities	Animals/Cell Models	Sample Preparation	Test Index	Effects	References
Treatment of HUA	SPF grade Kunming breed male mice, weighing 18 to 22 g. Oxygen oxazine acid potassium (50 mg/kg) + adenine (100 mg/kg) in sodium. Carboxymethylcellulose suspension for 21 days. CG, MG, LDG (1 g/kg), HDG (2 g/kg).	Dried and crushed, passed through 120 mesh sieve, 200 g of dry powder was placed in a distillation flask; 500 mL of distilled water was added and extracted 3 times at 85 °C under reflux, cooled and filtered, concentrated, and lyophilized.	The levels of UA, Cr and BUN, XO, TNF-α, and IL-1β. Lesions of renal tissue.	↓ XOD activity and UA levels (*n* = 10, *p* < 0.01 for 2 mg/kg). The aqueous extract at 2 mg/kg was effective in reducing the damage to the kidney structures.	[12]
Protection against RIRI	SPF level SD male rats, body weight 180 to 220 g. CG, MG, LDG (0.5 g/kg), HDG (1 g/kg).	Dried and crushed, passed through 120 mesh sieve, 200 g of dry powder was placed in a distillation flask; 500 mL of distilled water was added and extracted 3 times at 85 °C under reflux, cooled and filtered, concentrated, and lyophilized.	Cr, BUN, MDA, SOD activity, XOD activity, morphological lesions of renal tissue.	↓ The levels of MDA, Cr, and BUN and the activity of XOD (*p* < 0.05 for 1g/kg). ↑ SOD level (*p* < 0.01 for 1g/kg). The antioxidant capacity of the body was improved. The damage of oxidative stress was reduced, and the renal function after RIRI was improved.	[13]
Anti-fatigue	Balb/c mice, male, body mass 18–22 g, SPF grade. CG (saline), LDG (125 mg/kg), MDG (250 mg/kg), HDG (500 mg/kg). Mice in each group were weighed daily and administered by gavage at a dose of 0.1 mL/10 g for 14 days.	Take 1000 g of DJF, add 10 L of distilled water, soak for 2 h, then perform heat reflux extraction at 100 °C for 60 min, filter, concentrate and lyophilize.	Body weight, organ index, and time to exhaustion in weight-bearing swimming. Serum biochemical indexes: BLA, LDH, BUN levels. Liver glycogen, muscle glycogen, MDA, and SOD activity in liver tissue.	There was no significant effect (*p* > 0.05) on the body weight and organs of mice. The low, medium, and high doses of the extract significantly increased the weight-bearing swimming time of mice (*p* < 0.01), with a certain dose-dependent effect, and improved the anti-fatigue tolerance of mice. ↓ The BLA content of mice in the three dose groups (*p* < 0.01). The BUN content of mice in the three dose groups decreased by 17.21%, 25.05% and 31.82%, respectively (*p* < 0.01). ↑The contents of hepatic glycogen and myoglycogen (*p* < 0.01). The antioxidant capacity of mice was improved: ↓ MDA (*p* < 0.01), ↑ SOD level (*p* < 0.01).	[14]
Memory improvement	SPF grade Kunming breed mice. CG, MG, LDG (0.70 g/kg·bw), MDG (1.40 g/kg·bw), HDG (2.80 g/kg·bw).	Aqueous bath extraction: 1:20 (g/mL), 85 °C, 1 h, filtration, concentration, and lyophilization.	Open field experiment: record the total number of squares crossed and the number of hind limbs erected by the mice within 5 min. Automatic recording of the water maze experiment: recording the latency of escape in mice. TC, TG, HDL-C, LDL-C.	Improvement of exercise and memory: the number of squares traversed and the number of upright hind limbs were significantly increased in the HDG compared with the CG (*p* < 0.05). The latency of escape was significantly lower in the LDG, MDG, and HDG compared with the MG (*p* < 0.05). Regulation of blood lipids in mice: HDL-C levels were significantly higher in the LDG, MDG, and HDG compared to the CG (*p* < 0.05).	[15]
Anti-cervical cancer activity	SPF grade Kunming mice, weight 24 ± 2 g. Cervical cancer U27 cell line. Dose: 0.8624 g·kg^−1^, 0.2 mL·10 g^−1^.	Take 0.3 kg of dry powder and add 10 times the dose of distilled water to soak for 1 h, then water decoct 2 times and rotary evaporate to obtain the infusion.	Tumor inhibition rate.	Tumor suppression rate was 48.8%. Large necrosis of tumor cells, the disappearance of nuclei, and interstitial edema.	[16]
Anti-tumor	HCT-116 human rectal colon cancer cells. Dose: 50 μg/mL–400 μg/mL.	N/A	N/A	Endoplasmic reticulum stress induces apoptosis in cancer cells.	[16]

“↓” represents decrease or weakening. “↑” represents raising or strengthening. CG, normal group. LDG, low-dose group. MDA, malondialdehyde. SOD, superoxide dismutase. BLA, serum lactate. LDH, lactate dehydrogenase. TC, total cholesterol. TG, total triglycerides. HDL-C, high-density lipoprotein cholesterol. LDL-C, low density lipoprotein cholesterol.

**Table 3 foods-12-03379-t003:** Monomeric substances obtained by extraction of DJF with different volume fractions of ethanol.

Ethanol Volume Fraction/%	Extraction Site	Classification	Number/Species	Monomer Name	References
60	N/A	N/A	4	Quercitrin, naringenin, texifolin, isoquercitrin-6”-*O*-3′”,4′”,5′”-trihydroxybenzoyl.	[25]
		Diarylheptanes and their synthetic precursors	1	Juglanin D.	
		Phenolic acid	3	Gallic acid, ethyl gallate, protocatechuic acid.	
		Anthraquinones	2	4,5,8-trihydroxy-α-tetralone-5-*O*-β-*D*-glucopyranoside, 4,5-dihydroxy-α-tetralone-4-*O*-β-*D*-glucopyranoside.	
		Steroids	2	β-sitosterol, daucosterol.	
		Others	1	Taxifolin-3-*O*-α-l-arabinofuranoside.	
70	Ethyl acetate	Phenolic acid	3	Gallic acid, progallin A, protocatechuic acid.	[4,34,35,36,37,38]
		Flavonoids	28	Quercetin, dihydroquercetin, catechin, quercitrin, quercetin-3-*O*-(6”-galloyl)-β-D-galactopyranoside, kaempferol, naringenin, taxifolin-3-O-α-l-arabinofuranoside, (2S,3S)-taxifolin-3-*O*-α-*D*-arabinofuranoside, (2S,3S)-taxifolin-3-*O*-α-l-arabinofuranoside, Juglanoside A, Juglanoside B, Juglanoside E, 3,5,7-trihydroxylchromone-3-*O*-α-l-arabinofuranoside, taxifolin, taxifolin-3-β-d-xylopyranoside, (+)-catechin, catechin lactone A, naringenin derivatives naringenin 7-*O*-β-d-glucopyranoside, sakuranetin 5-*O*-β-d xylopyranoside, (2R)-eriodictyol-5-*O*-β-d-glucoside, 3-*O*-methylquercetin, avicularin, quercetin-3-*O*-α-d-arabinofuranoside, quercetin 3-*O*-β-d-xylopyranoside, quercetin-3-*O*-(6″-*O*-galloyl)-β-d-galactopyranoside, quercetin-3-*O*-β-d-glucopyranoside, luteolin.	
		Authraquinones	2	1,4,8-trihydroxy-3-naphthalenecarbox-ylicacid-1-*O*-β-*D*-glucopyranoside ethyl ester,(4S)-4-hydroxy-α-tetralone-4-*O*-β-*D*-(6′-*O*-4′-hydroxylbenzoyl)glucopyranoside.	
		Steroids	2	β-sitosterol, daucosterol.	
		Diarylheptanes and their synthetic precursors	1	Juglanin D.	
		Lignans	2	(+)-pinoresinol, (+)-syringaresinol.	
		Phenylbutanone glucoside	6	Juglandisde A, salviaplebeiaside, 3′-*O*-β-*D*-glucopyranoside of 4-(3,4′-dihydroxyphenyl)butan-2-one, benzyl-β-*D*-glucopyranoside, phenylethyl-β-*D*-glucopyranoside, methyl(6-*O*-p-hydroxy-benzoyl)-β-*D*-glucopyranoside.	
		Megastigmanes	11	Diamegastigmane A, diamegastigmane B, diamegastigmane C, blumenol B, vomifoliol, aglycone of euodionoside G, bridelionol C, myrsinionoside A, byzantionoside B, blumenol C glucoside, (6R, 9S)-6′-(4″-hydroxybenzoyl)-roseoside.	
		Phenylpropanoids	9	1-*O*-(Z)-coumaroyl, 6-*O*-(E)-coumaroyl-β-*D*-glucopyranoside, 1,6-di-*O*-(E)-coumaroyl-β-d-glucopyranoside, erythro-(7S,8R)-guaiacyl-glycerol-β-*O*-4′-dihydroconiferyl ether, 1-(4′-hydroxy-3′-methoxyphenyl)-2-[4″-(3-hydroxypropyl)-2″,6″-dimethoxyphenoxy]propane-1,3-diol, rosalaevin B, 5-methoxy-(+)-isolariciresinol, erythro-guaiacyl-glycerol-β-*O*-4′-(5′)-methoxylariciresinol, rhoiptelol B, dihydrodehydodiconiferyl alcohol.	
		Others	3	dihydrovomifoliol-9-*O*-β-*D*-glucopyranoside, bis(7-hydroxyheptyl) hexanedioate, 4-*O*-(2-hydroxymethylethyl)-dihydroconi-ferylalcohol.	
70	Petroleum ether	Flavonoids	1	Juglanin A.	[37]
		Steroids	1	β-sitosterol.	
70	N-butyl alcohol	Flavonoids	2	Quercetin-3-*O*-(4″-*O*-acetyl)-α-l-rhamnopyranoside, kaempferol-3-*O*-α-l-rhamnopyranoside.	[34]
		Phenolic acid	1	vanillin.	
70		Terpenoids	4	Juglansin A, juglansin B, juglansin C, juglansin D.	[23]
75	Ethyl acetate	Phenolic acid	10	Protocatechuic acid, vanillin, methyl gallate, vanillic acid, syringic acid, ellagic acid, gallic acid, ethyl gallate, p-hydroxybenzoic acid, juglanin D.	[24,39,40]
		Sesquiterpenoids	1	4′-dihydro-phaseic acid.	
		Authraquinones	3	Juglanoside E, emodin, isosclerone.	
		Steroids	1	oleanolic acid.	
		Others	3	Heptadecane, 3-hydroxy-1-(4-hydroxyphenyl)-1-propanone, glycerol-1-octadecanoate.	
95	Ethyl acetate	Phenolic acid	6	Gallic acid, dihydrophaseic acid, protocatechuic acid, p-hydroxybenzoic acid, vanillic acid, ethyl gallate.	[41]
		Megastigmanes	3	Blumenol B, (6R,9R)-9-hydroxymegastigman-4-en-3-one, (6R,9S)-9- hydroxymegastigman-4-en-3-one.	
		Flavonoids	3	Quercitrin, taxifolin-3-*O*-α-l-arabinofuranoside, dihydroquercetin.	
		Authraquinones	1	(4S)-4-hydroxy-1-tetralone.	
		Terpenoids	1	(+)-dehydrovomifoliol.	
95	Petroleum ether	Authraquinones	2	2-ethoxyjuglone, isosclerone.	[39]
		Flavonoids	1	Juglanin A.	
		Megastigmanes	1	4-megastigmen-3,9-dione.	
95	Water	Phenolic acid	4	3′-*O*-(E-4-coumaroyl)-quinic acid, dihydrophaseic acid, 5′-*O*-(E-4-coumaroyl)-quinic acid, vanillic acid-4-*O*-β-*D*-glucopyranoside.	[39]
		Others	1	Litchiol A.	
N/A	N/A	Phenolic acid	1	p-coumaric acid.	[19]
		Others	3	1′-methyl-2′-hydroxy) propane-*O*-α-*D*-glucopyranoside, (4′-hydroxyphenyl) methylene-*O*-β-*D*-glucopyranosyl-(4→1)-α-l-arabinopyranoside, 2-carboxy-5,7-dihydroxy3-naphthyl-β-*D*-glucopyranoside.	

**Table 7 foods-12-03379-t007:** Summary of DJF saponins extraction.

Extraction Conditions	Detection Method	Extraction Yield	Purification Conditions	References
95% ethanol, 80 °C reflux extraction, water-saturated n-butanol extraction.	UV-visible spectrophotometry (250 nm).	4.19%.	N/A	[107]
Material-to-liquid ratio of 1:25, 80% ethanol, 80 °C reflux extraction two times, each time 2 h.	UV-visible spectrophotometry (563 nm).	2.65%.	D101 macroporous adsorbent resin: maximum sample volume of 180 mL, water elution volume of 8 BV, volume fraction of eluent 50% ethanol, elution volume of 7 BV.	[108]
Ultrasonic-assisted technology: ethanol concentration of 60%, ultrasonic time of 3 min, material-to-liquid ratio of 1:25 (g/mL).	UV-visible spectrophotometry (460 nm).	1.86%.	N/A	[106]
Material-to-liquid ratio of 1:25, 80% ethanol, 80 °C reflux extraction two times, each time 2 h.	N/A	2.70%.	D101 macroporous adsorbent resin: maximum sample volume of 180 mL, water elution volume of 8 BV, volume fraction of eluent 50% ethanol, elution volume of 7 BV. The purity of total saponins increased from 27.40% to 50.80%, and the elution rate of total saponins could reach 82.40%.	[60]

## Data Availability

The data used to support the findings of this study can be made available by the corresponding author upon request.

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
