# Peer review of "A Review on Extracts, Chemical Composition and Product Development of Walnut Diaphragma Juglandis Fructus"

_foods, 2023, doi:10.3390/foods12183379_

Round 1

Reviewer 1 Report

The review highlights the significance of walnuts Diaphragma Juglandis Fructus (DJF). In recent years, this by-product has garnered attention due to its diverse chemical composition and bioactivities.

This review predominantly focused on aqueous and ethanol extracts of DJF.  Highlighting the gap in understanding the underlying mechanisms behind their bioactivities.

The review also points out that the bioactivities of polyphenols and polysaccharides within DJF.

Overall, this review provides insights into the existing research landscape concerning DJF, highlighting its bioactivities, chemical composition, and potential for diverse product development. Further exploration of its mechanisms and broader applications could lead to tapping into its full potential as a valuable natural resource.

Author Response

Thank the reviewer for your comments concerning our manuscript. The comments are valuable and helpful for our manuscript.

The study of Diaphragma Juglandis Fructus (DJF). is currently focused on the exploration of biological activity, with less research on the mechanisms of its activity. As you have commented, further exploration of its mechanisms and broader applications could lead to tapping into its full potential as a valuable natural resource. In the future, we will focus our research on the activity and mechanisms of DJF.

We hope to have another opportunity to communicate with you in the future. Thank you again for all the work you have done on our manuscript.

Thank you again for all the work you have done on our manuscript. Thank you for the valuable time you have spent on this manuscript.

Reviewer 2 Report

Comment and suggestion to authors:

 Manuscript ID: foods-2580108

Type: Review

Titled:  Extracts, chemical composition and product development of walnut Diaphragma Juglandis Fructus: A review

1)    Please, check and use the correct scientific name (legitimate name) of this plant in the manuscript.

 2)    Line 32, the authors wrote that “Walnut is a genus of Juglans and has a long history of cultivation in China,…..”. Generic name of walnut is “Juglans”. So, it should be revised to be “Walnut belongs to the genus Juglans, and has a long history of cultivation in China,…..”.

 3)    The figure 2 (Summary of biofunctions of different solvent extracts of DJF.) should be revised. The explanations in the photo should be revised to be easy to read and the supported references should be added.

4)    The figure 4 (Product development of DJF), the supported references should be added, if the authors did not prepare those photos by themselves. The authors should carefully check and follow the “Instructions for Authors” to prepare the figures and the other part of the manuscript.

5)    It would be also nice and interesting, if the authors can add the future perspective after the conclusion.

6)    The greater number of another related published works should be added to discuss with the results from this current study.

7)    There are some spelling mistakes and grammatical error found in this manuscript.

Comment and suggestion to authors:

Manuscript ID: foods-2580108

Type: Review

Titled:  Extracts, chemical composition and product development of walnut Diaphragma Juglandis Fructus: A review

1)    Please, check and use the correct scientific name (legitimate name) of this plant in the manuscript.

2)    Line 32, the authors wrote that “Walnut is a genus of Juglans and has a long history of cultivation in China,…..”. Generic name of walnut is “Juglans”. So, it should be revised to be “Walnut belongs to the genus Juglans, and has a long history of cultivation in China,…..”.

3)    The figure 2 (Summary of biofunctions of different solvent extracts of DJF.) should be revised. The explanations in the photo should be revised to be easy to read and the supported references should be added.

4)    The figure 4 (Product development of DJF), the supported references should be added, if the authors did not prepare those photos by themselves. The authors should carefully check and follow the “Instructions for Authors” to prepare the figures and the other part of the manuscript.

5)    It would be also nice and interesting, if the authors can add the future perspective after the conclusion.

6)    The greater number of another related published works should be added to discuss with the results from this current study.

7)    There are some spelling mistakes and grammatical error found in this manuscript.

Author Response

Thank the reviewer for your comments concerning our manuscript. The comments are valuable and helpful for improving our manuscript.

The revised manuscript and related comments are attached. We kindly ask you to review it.

Thank you again for all the work you have done on our manuscript. Thank you for the valuable time you have spent on this manuscript.

Reviewer 3 Report

The manuscript entitled “Extracts, chemical composition and product development of walnut Diaphragma Juglandis Fructus: A review” presented by Zhan et al., summaries a comprehensive review on walnut. Overall manuscript sounds good and delivering the scientific contents. However, some major exercise is required to improve further its content:   

·       Title need to be modified, as must not begin with extracts.

·       Please reform the sentence in Abstract Line 14-15 page 1

·       Please check the line in which claims are made on non-availability of any such studies (Line no- 17-18, Page 1 in Abstract Section). Please check these

https://pubmed.ncbi.nlm.nih.gov/37023838/

https://www.ncbi.nlm.nih.gov/pmc/articles/PMC9143591/

Please define the gap, how your study is different from these

·       In Introduction there is need to define past literature on different sources of walnut. Country wise production and effect on economy through import and export.

·       In introduction traditional and ethnobotanical application of walnut as food and medicine need to be discussed.

·       Please provide separate list of abbreviation

·       Discussion can be improved and the findings must be discussed with similar findings in details.

·       Any analytical study may add value.

·       Any section related to different areas of product in the market may add value (May be in table)

·       Any other future studies that need to warranted must be concluded.

·       Language and any other typological mistake can be address

·       Please check pattern of reference as per format.

Author Response

(The authors gave the same response as above.)

Reviewer 4 Report

This work reviews extracts, chemical compositions, and products of Diaphragma Juglandis Fructus. However, there are some points that need to be revised:

1. It would be better if the author focused on "extraction methods" rather than "extracts".

2. Figure 1: Compounds and bioactivity should be separated into different lines or columns.

3. Figure 2: It would be more effective to display each part of the walnut for each extraction solvent and mechanism.

4. Table 2: Ethanol concentration and extraction site should be included. If not available, please indicate as "N/A".

5. Line 367: Please verify "0H" – it should be "OH" not zero-H.

6. Line 427: Please correct "saponine" to "saponins".

7. Table 6's legend: The scientific name should be italicized.

Author Response

(The authors gave the same response as above.)

Round 2

Reviewer 2 Report

The revised version is improved, and can be accepted for publication.

Minor editing of English language required

Reviewer 4 Report

All comments are answered and addressed. It can be accepted to publish in this version.